# Simultaneous Synthesis of Vitamins D_2_, D_4_, D_5_, D_6_, and D_7_ from Commercially Available Phytosterol, β-Sitosterol, and Identification of Each Vitamin D by HSQC NMR

**DOI:** 10.3390/metabo9060107

**Published:** 2019-06-06

**Authors:** Shiro Komba, Eiichi Kotake-Nara, Wakako Tsuzuki

**Affiliations:** Food Component Analysis Unit, Food Research Institute, National Agriculture and Food Research Organization, 2-1-12, Kannondai, Tsukuba, Ibaraki 305-8642, Japan; ekotake@affrc.go.jp (E.K.-N.); wakako@affrc.go.jp (W.T.)

**Keywords:** vitamin D, simultaneous synthesis, commercial β-sitosterol, HSQC NMR

## Abstract

We succeeded in simultaneously synthesizing the vitamin D family, vitamins D_2_, D_4_, D_5_, D_6_, and D_7_, from β-sitosterol, which is sold as a commercially available reagent from Tokyo Chemical Industry Co., Ltd. It is officially sold as a mixture of four phytosterols {β-sitosterol (40–45%), campesterol (20–30%), stigmasterol, and brassicasterol}. Owing to this, we anticipated that, using this reagent, various vitamin D analogs could be synthesized simultaneously. We also synthesized vitamin D_3_ from pure cholesterol and analyzed and compared all vitamin D analogs (D_2_, D_3_, D_4_, D_5_, D_6_, and D_7_) by HSQC NMR. We succeeded in clearly demonstrating the difference in the NMR chemical shifts for each vitamin D analog.

## 1. Introduction

The vitamin D family [1] is known as vitamins D_2_ {ergocalciferol or (5*Z*,7*E*,22*E*)-(3*S*)-9,10-seco-5,7,10(19),22-ergostatetraen-3-ol}, D_3_ {cholecalciferol or (5*Z*,7*E*)-(3*S*)-9,10-seco-5,7,10(19)-cholestatrien-3-ol}, D_4_ {22,23-dihydroercalciol or (5*Z*,7*E*)-(3*S*)-9,10-seco-5,7,10(19)-ergostatrien-3-ol), D_5_ {(5*Z*,7*E*)-(3*S*)-9,10-seco-5,7,10(19)-stigmastatrien-3-ol}, D_6_ {(22*E*)-(24*R*)-ethyl-22,23-didehydrocalciol or (5*Z*,7*E*,22*E*)-(3*S*)-9,10-seco-5,7,10(19),22-stigmastatetraen-3-ol}, and D_7_ {(5*Z*,7*E*,24*R*)-(3*S*)-9,10-seco-5,7,10(19)-ergostatrien-3-ol} and contributes to important biological functions after converting to active forms [2]. Although these vitamin D analogs differ only in the structure of their side chains, they have different activities in vivo. For example, antirachitic activity using vitamins D_2_, D_3_, D_4_, D_5_, D_6_, and D_7_ showed a relative effect of 100:100:75:2:1:10, respectively [2,3]. In addition, active forms of vitamin D_3_ are used as anti-osteoporosis drugs, but hypercalcemia due to excessive intake has been reported as a side effect. Active forms of vitamins D_4_ and D_7_ have been reported to be effective in improving this side effect [4]. Therefore, it is very important to compare each vitamin D to examine its functions. Although methods for synthesizing each vitamin D analog from phytosterols have been established [5,6,7,8,9,10], it is necessary to obtain all vitamin D analogs using a simple method in order to compare and analyze the biological functions of each vitamin D analog in detail. However, methods for obtaining each vitamin D analog simultaneously have not yet been developed. Focusing on commercially available β-sitosterol (Figure 1, Compound 1), it contains not only campesterol but also stigmasterol and brassicasterol according to the manufacturer; therefore, we anticipated that four types of vitamin D could be synthesized simultaneously. Furthermore, the characteristic peaks of each vitamin D analog (vitamins D_2_, D_3_, D_4_, D_5_, D_6_, and D_7_) were examined by comparative analysis using HSQC NMR, and a simple identification method was examined for each vitamin D analog using HSQC NMR.

## 2. Results and Discussion

### 2.1. Simultanous Synthesis of Vitamin D Analogs

Commercially available β-sitosterol was chosen as the starting material to synthesize various vitamin D analogs. It is not a synthetic product, but an extract from nature, and is a mixture of campesterol, stigmasterol, and brassicasterol according to the manufacturer (Figure 1). It was thought that vitamins D_5_, D_7_, D_6_, and D_2_ could be synthesized using these four starting materials, respectively. The conversion of phytosterol to vitamin D was performed with reference to previously established conditions (Figure 2) [11,12]. First, the 3-position hydroxy group of commercially available β-sitosterol (**1**) was protected with an acetyl group (**2**), then it was brominated using *N*-bromosuccinimide (NBS), and 5,7-diene (**3**) was formed by debromination using tetrabutylammonium fluoride. Subsequently, the acetyl group at the 3-position hydroxy group was deprotected to synthesize provitamin D (**4**). It was converted to previtamin D (**5**) by irradiating with 280 nm UV light. Finally, previtamin D (**5**) was converted to vitamin D (**6**–**9**) by heating at 100 °C for 1 h. First, the synthesized vitamin D analogs (**6**–**9**) were separated from the raw material, previtamin D (**5**), using silica-gel column chromatography (ethyl acetate/*n*-hexane = 3:7). At this time, previtamin D (**5**) was not recovered. Next, in order to separate the synthesized vitamin D analogs, separation by HPLC was examined. When a 20 × 250 mm reverse phase C-18 HPLC column was used as a separation column and acetonitrile was used as a solvent, three large peaks were successfully separated. As a result of NMR and MS measurements, vitamin D_2_ (**9**, 1.75 mg, 67.6 min), a mixture (11.0 mg, 78.4 min) of vitamin D analogs, and vitamin D_5_ (**6**, 11.3 mg, 88.9 min) were found in ascending order of retention time (Figure 3A). The NMR spectra of vitamin D_2_ (**9**) with five methyl groups {Me-18 (^1^H *δ* = 0.55, ^13^C *δ* = 12.3 ppm), Me-21 (^1^H *δ* = 1.02, ^13^C *δ* = 21.1 ppm), Me-24^1^ (^1^H *δ* = 0.92, ^13^C *δ* = 17.6 ppm), Me-26 and Me-27 (^1^H *δ* = 0.82, 0.84, ^13^C *δ* = 19.6, 19.9 ppm)}, four methine groups {CH-6 (^1^H *δ* = 6.23, ^13^C *δ* = 122.5 ppm), CH-7 (^1^H *δ* = 6.03, ^13^C *δ* = 117.5 ppm), CH-22 (^1^H *δ* = 5.19, ^13^C *δ* = 135.6 ppm), CH-23 (^1^H *δ* = 5.19, ^13^C *δ* = 132.0 ppm)}, one methylidene group {CH_2_-19 (^1^H *δ* = 4.81, 5.04, ^13^C *δ* = 112.4 ppm)}, and a characteristic side chain {^13^C-20 (*δ* = 40.37 or 40.41 ppm), ^13^C-24 (*δ* = 42.8 ppm), ^13^C-25 (*δ* = 33.1 ppm)} were in good agreement with the NMR data of the compound reported by both Koszewski et al. [13] and Tsukida et al. [14]. The NMR spectra of vitamin D_5_ (**6**) with five methyl groups {Me-18 (^1^H *δ* = 0.55, ^13^C *δ* = 12.0 ppm), Me-21 (^1^H *δ* = 0.93, ^13^C *δ* = 18.9 ppm), Me-24^2^ (^1^H *δ* = 0.85, ^13^C *δ* = 12.0 ppm), Me-26 and Me-27 (^1^H *δ* = 0.82, 0.84, ^13^C *δ* = 19.0, 19.8 ppm)}, two methine groups {CH-6 (^1^H *δ* = 6.24, ^13^C *δ* = 122.5 ppm), CH-7 (^1^H *δ* = 6.04, ^13^C *δ* = 117.5 ppm)}, one methylidene group {CH_2_-19 (^1^H *δ* = 4.83, 5.05, ^13^C *δ* = 112.4 ppm)}, and a characteristic side chain {^13^C-20 (*δ* = 36.5 ppm), ^13^C-22 (*δ* = 33.9 ppm), ^13^C-23 (*δ* = 26.1 ppm), ^13^C-24 (*δ* = 45.84 or 45.86 or 45.93 ppm), ^13^C-24^1^ (*δ* = 23.1 ppm), ^13^C-25 (*δ* = 29.0 or 29.2 ppm)} were in good agreement with the NMR data of the compound reported by Napoli et al. [15].

In order to separate the mixture obtained here, HPLC purification studies were carried out under various solvent conditions using a C-18 HPLC column; however, it was not possible to find solvent conditions that could be separated the mixture effectively. Therefore, when COSMOSIL Cholester (Nacalai Tesque Inc, 4.6 × 250 mm) with cholesterol residues on the surface was used as an HPLC column and acetonitrile was used as a solvent, three large peaks were successfully separated. Due to the small diameter of this column, all mixtures were separated over 20 iterations. As a result of NMR and MS measurements, vitamin D_6_ (**8**, 2.41 mg, 22.9 min), vitamin D_7_ (**7**, 3.52 mg, 24.9 min), and vitamin D_4_ (**10**, 1.85 mg, 26.4 min) were found in ascending order of retention time (Figure 3B).

At the beginning, we did not expect to be able to successfully synthesize vitamin D_4_. The fact that vitamin D_4_ was synthesized indicates that 22,23-dihydrobrassicasterol was contained in the starting compound, commercially available β-sitosterol (Figure 4). The yield of the final step was 39%. It was calculated from the total weight (20.8 mg) of the obtained vitamin D analogs, including vitamin D_4_. Compound **5** (53.31 mg) was reacted to obtain vitamin D_2_ (**9**: 1.75 mg, 4.41 μmol), vitamin D_4_ (**10**: 1.85 mg, 4.64 μmol), vitamin D_5_ (**6**: 11.27 mg, 27.3 μmol), vitamin D_6_ (**8**: 2.41 mg, 5.87 μmol), and vitamin D_7_ (**7**: 3.52 mg, 8.83 μmol). Assuming that the ratio of brassicasterol/22,23-dihydrobrassicasterol/β-sitosterol/stigmasterol/campesterol contained in commercially available β-sitosterol does not change in each reaction, the weight percentage of each compound in commercially available β-sitosterol is 1.75:1.85:11.27:2.41:3.52 = 8.4 wt%/8.9 wt%/54.2 wt%/11.6 wt%/16.9 wt%, respectively. The molar ratio is 4.41:4.64:27.3:5.87:8.83= 8.6%/9.1%/53.5%/11.5%/17.3%, respectively. The NMR spectra of vitamin D_6_ (**8**) with five methyl groups {Me-18 (^1^H *δ* = 0.56, ^13^C *δ* = 12.2 ppm), Me-21 (^1^H *δ* = 1.02, ^13^C *δ* = 21.3 ppm), Me-24^2^ (^1^H *δ* = 0.81, ^13^C *δ* = 12.2 ppm), Me-26 and Me-27 (^1^H *δ* = 0.79, 0.85, ^13^C *δ* = 19.0, 21.1 ppm)}, four methine groups {CH-6 (^1^H *δ* = 6.23, ^13^C *δ* = 122.5 ppm), CH-7 (^1^H *δ* = 6.03, ^13^C *δ* = 117.5 ppm), CH-22 (^1^H *δ* = 5.17, ^13^C *δ* = 138.1 ppm), CH-23 (^1^H *δ* = 5.02, ^13^C *δ* = 129.5 ppm)}, one methylidene group {CH_2_-19 (^1^H *δ* = 4.81, 5.04, ^13^C *δ* = 112.4 ppm)}, and a characteristic side chain {^13^C-20 (*δ* = 40.7 ppm), ^13^C-24 (*δ* = 51.2 ppm), ^13^C-24^1^ (*δ* = 25.4 ppm), ^13^C-25 (*δ* = 31.9 ppm)} were in good agreement with the NMR data of the side chain of the stigmasterol [16,17]. The NMR spectra of vitamin D_7_ (**7**) with five methyl groups {Me-18 (^1^H *δ* = 0.54, ^13^C *δ* = 12.0 ppm), Me-21 (^1^H *δ* = 0.92, ^13^C *δ* = 18.8 ppm), Me-24^1^ (^1^H *δ* = 0.78, ^13^C *δ* = 15.4 ppm), Me-26 and Me-27 (^1^H *δ* = 0.81, 0.85, ^13^C *δ* = 18.2, 20.2 ppm)}, two methine groups {CH-6 (^1^H *δ* = 6.24, ^13^C *δ* = 122.5 ppm), CH-7 (^1^H *δ* = 6.04, ^13^C *δ* = 117.5 ppm)}, one methylidene group {CH_2_-19 (^1^H *δ* = 4.82, 5.05, ^13^C *δ* = 112.4 ppm)}, and a characteristic side chain {^13^C-20 (*δ* = 36.2 ppm), ^13^C-22 (*δ* = 33.7 ppm), ^13^C-23 (*δ* = 30.3 ppm), ^13^C-24 (*δ* = 38.8 ppm), ^13^C-25 (*δ* = 32.4 ppm)} were in good agreement with the NMR data of the side chain of the campesterol [17,18,19]. The NMR spectra of vitamin D_4_ (**10**) with five methyl groups {Me-18 (^1^H *δ* = 0.54, ^13^C *δ* = 12.0 ppm), Me-21 (^1^H *δ* = 0.92, ^13^C *δ* = 19.0 ppm), Me-24^1^ (^1^H *δ* = 0.78, ^13^C *δ* = 15.4 ppm), Me-26 and Me-27 (^1^H *δ* = 0.79, 0.85, ^13^C *δ* = 17.6, 20.5 ppm)}, two methine groups {CH-6 (^1^H *δ* = 6.23, ^13^C *δ* = 122.5 ppm), CH-7 (^1^H *δ* = 6.03, ^13^C *δ* = 117.5 ppm)}, one methylidene group {CH_2_-19 (^1^H *δ* = 4.82, 5.05, ^13^C *δ* = 112.4 ppm)}, and a characteristic side chain {^13^C-20 (*δ* = 36.5 ppm), ^13^C-22 (*δ* = 33.7 ppm), ^13^C-23 (*δ* = 30.6 ppm), ^13^C-24 (*δ* = 39.1 ppm), ^13^C-25 (*δ* = 31.5 ppm)} were in good agreement with the NMR data of the side chain of the 22,23-dihydrobrassicasterol (**11**) [17,20]. All carbon-13 chemical shifts are shown in Table 1 in order to elucidate the characteristic chemical shifts of each vitamin D analog.

### 2.2. Comparison of Each Vitamin D Analog by HSQC NMR

Since vitamin D analogs D_2_, D_4_, D_5_, D_6_, and D_7_ were synthesized from commercially available β-sitosterol, the remaining vitamin D_3_ was synthesized from cholesterol in the same manner as in the above synthesis method (Figure 5, Appendix A and the section of “The experimental details on the synthesis of vitamin D_3_ (**13**)” in the Appendix A). Vitamin D_3_ is commercially available and does not need to be synthesized this time. However, in order to reconfirm the effectiveness of this synthetic method, it was intentionally synthesized from cholesterol. As pure cholesterol was used as the starting material, all purification was carried out using silica-gel column chromatography and vitamin D_3_ could be synthesized without HPLC purifications. The NMR spectra of vitamin D_3_ (**13**) with four methyl groups {Me-18 (^1^H *δ* = 0.54, ^13^C *δ* = 12.0 ppm), Me-21 (^1^H *δ* = 0.92, ^13^C *δ* = 18.8 ppm), Me-26 and Me-27 (^1^H *δ* = 0.86, 0.87, ^13^C *δ* = 22.5, 22.8 ppm)}, two methine groups {CH-6 (^1^H *δ* = 6.23, ^13^C *δ* = 122.5 ppm), CH-7 (^1^H *δ* = 6.03, ^13^C *δ* = 117.5 ppm)}, one methylidene group {CH_2_-19 (^1^H *δ* = 4.82, 5.05, ^13^C *δ* = 112.4 ppm)}, and a characteristic side chain {^13^C-20 (*δ* = 36.1 ppm), ^13^C-22 (*δ* = 36.1 ppm), ^13^C-23 (*δ* = 23.9 ppm), ^13^C-24 (*δ* = 39.5 ppm), ^13^C-25 (*δ* = 28.0 ppm)} were in good agreement with the NMR data of vitamin D_3_ reported by Kruk et al. [21]. The HSQC NMR spectra of all the vitamin D analogs obtained were compared (Figure 6 and Figure 7A). As expected, all the vitamin D analogs have different structures of their side chain only; therefore, the NMR chemical shifts of the side chains showed unique values. The following characteristic peaks were observed. Vitamin D_2_ has CH-24, 25, vitamin D_3_ has CH-22, 23, 24, 25, 26, 27, vitamin D_4_ has CH-22, 23, 25, vitamin D_5_ has CH-24, 24^1^, 24^2^, vitamin D_6_ has CH-24, 24^1^, 24^2^, 25, and vitamin D_7_ has CH-23, 25. Although not in Figure 7A, both vitamins D_2_ and D_6_ also have CH-22, 23 as characteristic peaks. Thus, the HSQC NMR of the mixture was measured since the characteristic peaks of each vitamin D analog were identified by HSQC NMR, and it was verified whether or not it was possible to determine what kind of vitamin D analog was contained. We decided to use the mixture of vitamin D that was synthesized in this study. These were the final products separated from unreacted raw materials by silica-gel column chromatography and contained all the vitamin D analogs, except for vitamin D_3_. As a result of HSQC NMR, characteristic peaks of each vitamin D analog could be identified (Figure 6B). The identified characteristic peaks of each vitamin D analog were as follows; vitamin D_2_ has CH-24, vitamin D_3_ has no characteristic peaks, vitamin D_4_ has CH-23, vitamin D_5_ has CH-24, 24^1^, vitamin D_6_ has CH-24, 24^1^, and vitamin D_7_ has CH-23. From this analysis, it was found that vitamin D_2_, D_4_, D_5_, D_6_, and D_7_ are contained and vitamin D_3_ is not contained. Thus, HSQC NMR can be used to quickly identify which of the vitamin D analogs is contained in the mixture.

## 3. Materials and Methods 

### 3.1. Reagents and Conditions

^1^H NMR and ^13^C NMR spectra were obtained in CDCl_3_ on a Bruker BioSpin spectrometer (AV 400, Bruker Corporation, Madison, MA, USA). Chemical shifts are given in ppm and referenced to Me_4_Si (δ 0.00). The following abbreviations are used for the characterization of NMR signals: s = singlet, d = doublet, t = triplet, m = multiplet. ESI-Orbitrap-MS spectra were recorded on a Thermo Fisher Scientific instrument (VELOS PRO, Thermo Fisher Scientific Inc., Waltham, MA, USA). The optical rotations were determined in chloroform on a Jasco instrument (P-1020-GT, JASCO Corporation, Tokyo, Japan) under ambient temperature. UV irradiation was performed by combining a Ushio device UV lamp (SX-UID 501MAMQQ, Ushio Inc., Tokyo, Japan) and a Bunkoukeiki device monochromator (UB-100KC, Bunkoukeiki Co.,Ltd., Tokyo, Japan). HPLC condition 1: Reverse-phase HPLC separation of target compounds (vitamins D_2_ and D_5_) was performed using a Waters HPLC system with a Mightysil RP-18 GP 250-20 column (20 × 250 mm, 5 μm; Kanto Chemical Co., Inc., Tokyo, Japan) and detected at 254 nm. Initially, acetonitrile, at a flow rate of 1 mL/min, was used. Subsequently, the flow rate was increased to 5 mL/min over 1 min, and this condition was maintained. HPLC condition 2: For separation of vitamins D_4_, D_6_, and D_7_, reverse-phase HPLC separation was performed using a COSMOSIL Cholester column (4.6 × 250 mm; Nacalai Tesque Inc., Kyoto, Japan) and detected at 254 nm. Acetonitrile, at a flow rate of 1 mL/min, was used and this condition was maintained. All the reagents and solvents used were reagent grade.

### 3.2. Simultaneous Synthesis of Vitamin D Analogs, Vitamins D_2_ (**9**), D_4_ (**10**), D_5_ (**6**), D_6_ (**8**), and D_7_ (**7**).

β-Sitosterol **1** (contains campesterol, stigmasterol, and brassicasterol according to the manufacturer) was purchased from Tokyo Chemical Industry Co., Ltd. (catalog number: S0040, 25 g, JPY 4900) (3.0 g, 7.23 mmol calculated as β-sitosterol) was dissolved in pyridine (20 mL), acetic anhydride (5.0 mL, 52.9 mmol) was added, and the mixture was stirred at 45 °C for 15 h. After ice was added, the mixture was stirred for 1 h, extracted with chloroform, washed with 2N aq. HCl, concentrated, and then the main products were crystallized with ethanol to obtain compound **2** (3.24 g, 98%, single spot on TLC). Compound **2** (3.0 g, 6.57 mmol calculated as a β-sitosterol derivative) was dissolved in cyclohexane (80 mL) at 65 °C, *N*-bromosuccinimide (NBS; 1.75 g, 9.83 mmol) was added and then stirred under reflux conditions (90 °C) for 1 h. After cooling the reaction to room temperature, water (100 mL) was added and then stirred for 1 h. The mixture was extracted with *n*-hexane, washed with water, then concentrated and dried in vacuo. To the resulting mixture, a 1.0 M solution of tetrabutylammonium fluoride in THF (9.9 mL) was added and stirred at room temperature for 12 h. The reaction product was extracted using *n*-hexane, washed with water, concentrated, and then the main products were separated with silica-gel column chromatography (ethyl acetate/*n*-hexane 1:10) to obtain compound **3** (1.36 g, 45%, single spot on TLC). Compound **3** (1.29 g, 2.84 mmol calculated as β-sitosterol derivative) was dissolved in dichloromethane (8 mL) and methanol (20 mL), 28% NaOMe in MeOH was added until pH 10 and then stirred for 2 h. After the reaction mixture was concentrated, the main products were separated with silica-gel column chromatography (ethyl acetate/*n*-hexane 3:7) to obtain compound **4** (0.900 g, 77%, single spot on TLC). Compound **4** (100 mg, 0.242 mmol calculated as β-sitosterol derivative) was dissolved in 0.1% 3-*tert*-butyl-4-hydroxyanisole (BHA) in cyclohexane (10 mL) and transferred to a petri dish. While stirring the mixture in a petri dish covered with a polyvinylidene chloride food wrap, the mixture was irradiated with UV at 280 nm (9.03 mW/cm^2^) for 4 h. After the reaction mixture was concentrated, the main products were separated using silica-gel column chromatography (ethyl acetate/*n*-hexane 3:7) to obtain compound **5** (28 mg, 28%, single spot on TLC). Compound **5** (53.31 mg, 0.129 mmol calculated as β-sitosterol derivative) was dissolved in 0.1% BHA in cyclohexane (20 mL) and the mixture was stirred under reflux conditions (100 °C) for 1 h. After the reaction mixture was concentrated, the main products were separated using silica-gel column chromatography (ethyl acetate/*n*-hexane 3:7) to obtain the vitamin D mixture (29.4 mg, 55%, single spot on TLC). The obtained vitamin D mixture was separated by HPLC condition 1 to obtain pure vitamin D_2_ (**9**; 1.75 mg, retention time 67.6 min) and vitamin D_5_ (**6**; 11.3 mg, retention time 88.9 min). Other vitamin D analogs were not separated by this condition and a single peak was obtained (11.0 mg, retention time 78.4 min). The obtained vitamin D mixture was further separated by HPLC condition 2 to obtain pure vitamin D_4_ (**10**; 1.85 mg, retention time 26.4 min), vitamin D_6_ (**8**; 2.41 mg, retention time 22.9 min), and vitamin D_7_ (**7**; 3.52 mg, retention time 24.9 min). Vitamin D_2_ (**9**); [α]${}_{D}^{25}$ = +34.3 (*c* = 0.088, chloroform); ^1^H-NMR (400 MHz, CDCl_3_), *δ* = 0.55 (s, 3H, Me-18), 0.82 (d, 3H, *J* = 6.4 Hz, Me-26 or Me-27), 0.84 (d, 3H, *J* = 6.6 Hz, Me-26 or Me-27), 0.92 (d, 3H, *J* = 6.8 Hz, Me-24^1^), 1.02 (d, 3H, *J* = 6.7 Hz, Me-21), 1.26–1.35 (m, 3H, H-12a, H-16a, H-17), 1.43–1.57 (m, 4H, H-11a, H-15a, H-15b, H-25), 1.64–1.73 (m, 4H, H-2a, H-9a, H-11b, H-16b), 1.85 (m, 1H, H-24), 1.89–2.05 (m, 3H, H-2b, H-14, H-20), 2.18 (ddd, 1H, *J* = 13.1, 8.1, 4.2 Hz, H-1a), 2.28 (dd, 1H, *J* = 13.2, 7.4 Hz, H-4a), 2.40 (ddd, 1H, *J* = 13.1, 7.5, 4.6 Hz, H-1b), 2.57 (dd, 1H, *J* = 13.1, 3.2 Hz, H-4b), 2.82 (m, 1H, H-9b), 3.95 (m, 1H, H-3), 4.81 (d, 1H, *J* = 2.2 Hz, H-19a), 5.04 (1H, H-19b), 5.19 (m, 2H, H-22, H-23), 6.03 (d, 1H, *J* = 11.3 Hz, H-7), 6.23 (d, 1H, *J* = 11.2 Hz, H-6); ^13^C-NMR (100 MHz, CDCl_3_), *δ* = 12.3 (C-18), 17.6 (C-24^1^), 19.6 (C-26 or C-27), 19.9 (C-26 or C-27), 21.1 (C-21), 22.2 (C-15), 23.6 (C-11), 27.8 (C-16), 29.0 (C-9), 31.9 (C-1), 33.1 (C-25), 35.2 (C-2), 40.37 (C-12 or C-20), 40.41 (C-12 or C-20), 42.8 (C-24), 45.8 (C-13), 45.9 (C-4), 56.4 (C-14 or C17), 56.5 (C-14 or C-17), 69.2 (C-3), 112.4 (C-19), 117.5 (C-7), 122.5 (C-6), 132.0 (C-23), 135.0 (C-5), 135.6 (C-22), 142.2 (C-8), 145.1 (C-10); ESI-Orbitrap-MS, calcd. for C_28_H_45_O^+^ (M + H)^+^: 397.3465, found *m/z*: 397.3461. Vitamin D_4_ (**10**); [α]${}_{D}^{25}$ = +23.8 (*c* = 0.093, chloroform); ^1^H-NMR (400 MHz, CDCl_3_), *δ* = 0.54 (s, 3H, Me-18), 0.78 (d, 3H, *J* = 6.8 Hz, Me-24^1^), 0.79 (d, 3H, *J* = 6.8 Hz, Me-26 or Me-27), 0.85 (d, 3H, *J* = 6.8 Hz, Me-26 or Me-27), 0.92 (d, 3H, *J* = 6.1 Hz, Me-21), 0.93–0.97 (m, 2H, H-22a, H-23a), 1.21 (m, 1H, H-24), 1.25–1.41 (m, 6H, H-12a, H-16a, H-17, H-20, H-22b, H-23b), 1.45–1.60 (m, 4H, H-11a, H-15a, H-15b, H-25), 1.65–1.72 (m, 3H, H-2a, H-9a, H-11b), 1.84–2.01 (m, 4H, H-2b, H-12b, H-14, H-16b), 2.18 (ddd, 1H, *J* = 13.5, 8.5, 4.7 Hz, H-1a), 2.28 (dd, 1H, *J* = 13.1, 7.5 Hz, H-4a), 2.40 (ddd, 1H, *J* = 13.3, 7.5, 4.5 Hz, H-1b), 2.57 (dd, 1H, *J* = 13.3, 3.6 Hz, H-4b), 2.82 (m, 1H, H-9b), 3.95 (m, 1H, H-3), 4.82 (d, 1H, *J* =2.4 Hz, H-19a), 5.05 (m, 1H, H-19b), 6.03 (d, 1H, *J* = 11.3 Hz, H-7), 6.23 (d, 1H, *J* = 11.2 Hz, H-6); ^13^C-NMR (100 MHz, CDCl_3_), *δ* = 12.0 (C-18), 15.4 (C-24^1^), 17.6 (C-26 or C-27), 19.0 (C-21), 20.5 (C-26 or C-27), 22.3 (C-15), 23.6 (C-11), 27.6 (C-16), 29.0 (C-9), 30.6 (C-23), 31.5 (C-25), 31.9 (C-1), 33.7 (C-22), 35.2 (C-2), 36.5 (C-20), 39.1 (C-24), 40.5 (C-12), 45.85 (C-4 or C-13), 45.93 (C-4 or C-13), 56.3 (C-14), 56.5 (C-17), 69.2 (C-3), 112.4 (C-19), 117.5 (C-7), 122.5 (C-6), 135.0 (C-5), 142.4 (C-8), 145.1 (C-10); ESI-Orbitrap-MS, calcd. for C_28_H_47_O^+^ (M + H)^+^: 399.3621, found *m/z*: 399.3616. Vitamin D_5_ (**6**); [α]${}_{D}^{25}$ = +40.3 (*c* = 0.56, chloroform); ^1^H-NMR (400 MHz, CDCl_3_), *δ* = 0.55 (s, 3H, Me-18), 0.82 (d, 3H, *J* = 6.9 Hz, Me-26 or Me-27), 0.84 (d, 3H, *J* = 7.3 Hz, Me-26 or Me-27), 0.85 (t, 3H, *J* = 7.8, 7.8 Hz, Me-24^2^), 0.93 (m, 4H, Me-21, H-24), 1.03 (m, 1H, H-22a), 1.17 (m, 2H, H-23a, H-23b), 1.22–1.37 (m, 7H, H-12a, H-16a, H-17, H-20, H-22b, H-24^1^a, H-24^1^b), 1.45–1.58 (m, 3H, H-11a, H-15a, H-15b), 1.65–1.72 (m, 4H, H-2a, H-9a, H-11b, H-25), 1.85–2.02 (m, 4H, H-2b, H-12b, H-14, H-16b), 2.18 (ddd, 1H, *J* = 13.4, 8.4, 4.7 Hz, H-1a), 2.29 (dd, 1H, *J* = 13.1, 7.5 Hz, H-4a), 2.40 (ddd, 1H, *J* = 13.1, 7.7, 4.8 Hz, H-1b), 2.58 (dd, 1H, *J* = 13.0, 3.3 Hz, H-4b), 2.82 (m, 1H, H-9b), 3.95 (m, 1H, H-3), 4.83 (broad s, 1H, H-19a), 5.05 (s, 1H, H-19b), 6.04 (d, 1H, *J* = 11.2 Hz, H-7), 6.24 (d, 1H, *J* = 11.2 Hz, H-6); ^13^C-NMR (100 MHz, CDCl_3_), *δ* = 12.0 (C-18 and C-24^2^), 18.9 (C-21), 19.0 (C-26 or C-27), 19.8 (C-26 or C-27), 22.3 (C-15), 23.1 (C-24^1^), 23.6 (C-11), 26.1 (C-23), 27.7 (C-16), 29.0 (C-9 or C-25), 29.2 (C-9 or C-25), 31.9 (C-1), 33.9 (C-22), 35.2 (C-2), 36.5 (C-20), 40.5 (C-12), 45.84 (C-4 or C-13 or C-24), 45.86 (C-4 or C-13 or C-24), 45.93 (C-4 or C-13 or C-24), 56.4 (C-14), 56.5 (C-17), 69.2 (C-3), 112.4 (C-19), 117.5 (C-7), 122.5 (C-6), 135.0 (C-5), 142.3 (C-8), 145.1 (C-10); ESI-Orbitrap-MS, calcd. for C_29_H_49_O^+^ (M + H)^+^: 413.3778, found *m/z*: 413.3778. Vitamin D_6_ (**8**); [α]${}_{D}^{25}$ = +37.3 (*c* = 0.12, chloroform); ^1^H-NMR (400 MHz, CDCl_3_), *δ* = 0.56 (s, 3H, Me-18), 0.79 (d, 3H, *J* = 6.6 Hz, Me-26 or Me-27), 0.81 (d, 3H, *J* = 7.4 Hz, Me-24^2^), 0.85 (d, 3H, *J* = 6.4 Hz, Me-26 or Me-27), 1.02 (d, 3H, *J* = 6.6 Hz, Me-21), 1.17 (m, 1H, H-24^1^a), 1.23–1.37 (m, 3H, H-12a, H-16a, H-17), 1.37–1.60 (m, 6H, H-11a, H-15a, H-15b, H-24^1^b, H-24, H-25), 1.64–1.79 (m, 4H, H-2a, H-9a, H-11b, H-16b), 1.92 (m, 1H, H-2b), 1.95–2.09 (m, 3H, H-12b, H-14, H-20), 2.18 (ddd, 1H, *J* = 13.2, 8.4, 4.7 Hz, H-1a), 2.28 (dd, 1H, *J* = 13.0, 7.4 Hz, H-4a), 2.40 (ddd, 1H, *J* = 13.3, 7.4, 4.7 Hz, H-1b), 2.57 (dd, 1H, *J* = 13.1, 3.6 Hz, H-4b), 2.82 (m, 1H, H-9b), 3.95 (m, 1H, H-3), 4.81 (d, 1H, *J* = 2.4 Hz, H-19a), 5.02 (dd, 1H, *J* = 15.4, 8.8 Hz, H-23), 5.04 (1H, H-19b), 5.17 (dd, 1H, *J* = 15.2, 8.6 Hz, H-22), 6.03 (d, 1H, *J* = 11.3 Hz, H-7), 6.23 (d, 1H, *J* = 11.2 Hz, H-6); ^13^C-NMR (100 MHz, CDCl_3_), *δ* = 12.2 (C-18 and C-24^2^), 19.0 (C-26 or C-27), 21.1 (C-26 or C-27), 21.3 (C-21), 22.3 (C-15), 23.6 (C-11), 25.4 (C-24^1^), 28.2 (C-16), 29.0 (C-9), 31.9 (C-1 and C-25), 35.2 (C-2), 40.4 (C-12), 40.7 (C-20), 45.8 (C-13), 45.9 (C-4), 51.2 (C-24), 56.4 (C-17), 56.5 (C-14), 69.2 (C-3), 112.4 (C-19), 117.5 (C-7), 122.5 (C-6), 129.5 (C-23), 135.1 (C-5), 138.1 (C-22), 142.2 (C-8), 145.1 (C-10); ESI-Orbitrap-MS, calcd. for C_29_H_47_O^+^ (M + H)^+^: 411.3621, found *m/z*: 411.3615. Vitamin D_7_ (**7**); [α]${}_{D}^{25}$ = +42.0 (*c* = 0.18, chloroform); ^1^H-NMR (400 MHz, CDCl_3_), *δ* = 0.54 (s, 3H, Me-18), 0.78 (d, 3H, *J* = 6.6 Hz, Me-24^1^), 0.81 (d, 3H, *J* = 6.8 Hz, Me-26 or Me-27), 0.85 (d, 3H, *J* = 6.8 Hz, Me-26 or Me-27), 0.92 (d, 3H, *J* = 6.3 Hz, Me-21), 1.02–1.16 (m, 2H, H-22a, H-23a), 1.16–1.36 (m, 7H, H-12a, H-16a, H-17, H-20, H-22b, H-23b, H-24), 1.45–1.58 (m, 4H, H-11a, H-15a, H-15b, H-25), 1.65–1.72 (m, 3H, H-2a, H-9a, H-11b), 1.85–2.02 (m, 4H, H-2b, H-12b, H-14, H-16b), 2.18 (ddd, 1H, *J* = 13.2, 8.4, 4.7 Hz, H-1a), 2.29 (dd, 1H, *J* = 13.1, 7.5 Hz, H-4a), 2.40 (ddd, 1H, *J* = 13.3, 7.9, 4.9 Hz, H-1b), 2.58 (dd, 1H, *J* = 13.2, 3.5 Hz, H-4b), 2.82 (m, 1H, H-9b), 3.95 (m, 1H, H-3), 4.82 (broad d, 1H, *J* = 2.2 Hz, H-19a), 5.05 (broad s, 1H, H-19b), 6.04 (d, 1H, *J* = 11.3 Hz, H-7), 6.24 (d, 1H, *J* = 11.3 Hz, H-6); ^13^C-NMR (100 MHz, CDCl_3_), *δ* = 12.0 (C-18), 15.4 (C-24^1^), 18.2 (C-26 or C-27), 18.8 (C-21), 20.2 (C-26 or C-27), 22.3 (C-15), 23.6 (C-11), 27.7 (C-16), 29.0 (C-9), 30.3 (C-23), 31.9 (C-1), 32.4 (C-25), 33.7 (C-22), 35.2 (C-2), 36.2 (C-20), 38.8 (C-24), 40.5 (C-12), 45.86 (C-4 or C-13), 45.93 (C-4 or C-13), 56.4 (C-14), 56.6 (C-17), 69.2 (C-3), 112.4 (C-19), 117.5 (C-7), 122.5 (C-6), 135.0 (C-5), 142.4 (C-8), 145.1 (C-10); ESI-Orbitrap-MS, calcd. for C_28_H_47_O^+^ (M + H)^+^: 399.3621, found *m/z*: 399.3614.

## 4. Conclusions

Using commercially available phytosterols, vitamins D_2_, D_4_, D_5_, D_6_, and D_7_ were simultaneously synthesized using conventional methods and were successfully isolated and purified. The plant-derived β-sitosterol sold as a reagent by Tokyo Chemical Industry Co., Ltd., is known that to contain campesterol, stigmasterol, and brassicasterol. Furthermore, 22,23-dihydrobrassicaseterol is also supposedly present. Therefore, this is a good resource for the synthesis of the vitamin D family.

For further analysis, we synthesized vitamin D_3_ from cholesterol and successfully obtained all the vitamin D analogs. NMR analysis indicated that characteristic peaks of each vitamin D analog can be observed using HSQC NMR. In addition, certain papers on vitamin D analogs have reported incorrect assignments of NMR chemical shifts [14,22]; therefore, by reporting the correct assignment of the vitamin D family here, this study can provide the basis for future vitamin D research. These data will be used to enable comprehensive NMR analyses of vitamin D analogs, and the compounds synthesized here will be used to elucidate the functions of each vitamin D analog in our future studies.

## Figures and Tables

**Figure 1 metabolites-09-00107-f001:**
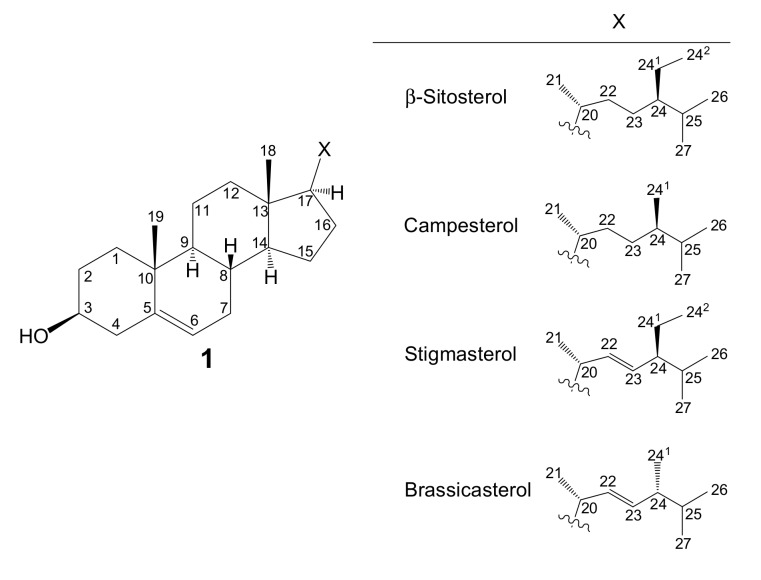
The structures of four phytosterols. β-Sitosterol **1** (catalog number: S0040, 25g, JPY 4900), sold by Tokyo Chemical Industry Co., Ltd. (Tokyo, Japan), is officially sold as a mixture of these four types of phytosterols.

**Figure 2 metabolites-09-00107-f002:**
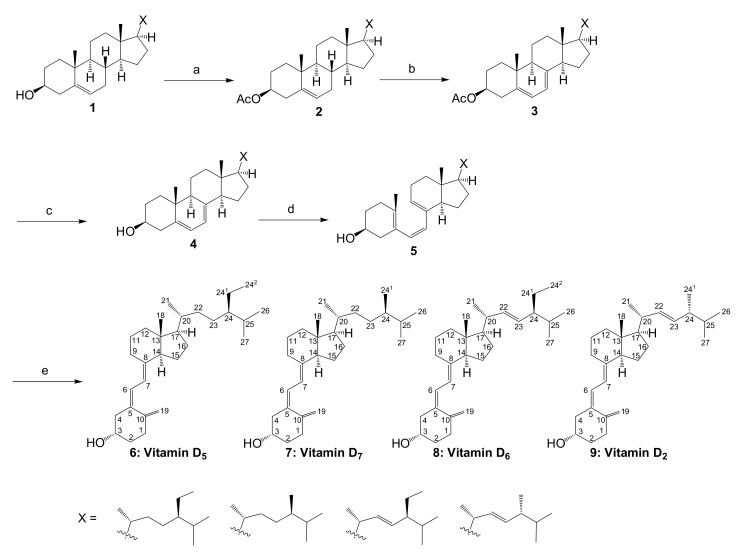
Reagents and conditions. (**a**) Ac_2_O, pyridine, 45 °C, 15 h, 98%. (**b**) (**1**) *N*-bromosuccinimide (NBS), cyclohexane, reflux, 1 h, (**2**) 1.0 M Bu_4_NF/THF, room temperature, 12 h, 45%. (**c**) 28% NaOMe in MeOH, CH_2_Cl_2_/MeOH, room temperature, 2 h, 77%. (**d**) 0.1% 3-*tert*-butyl-4-hydroxyanisole (BHA) in cyclohexane, 280 nm, 9.03 mW/cm^2^, room temperature, 4 h, 28%. (**e**) 0.1% BHA in cyclohexane, reflux, 1 h, 39% {It was calculated from the total weight (20.8 mg) of the obtained vitamin D analogs, including vitamin D_4_.}.

**Figure 3 metabolites-09-00107-f003:**
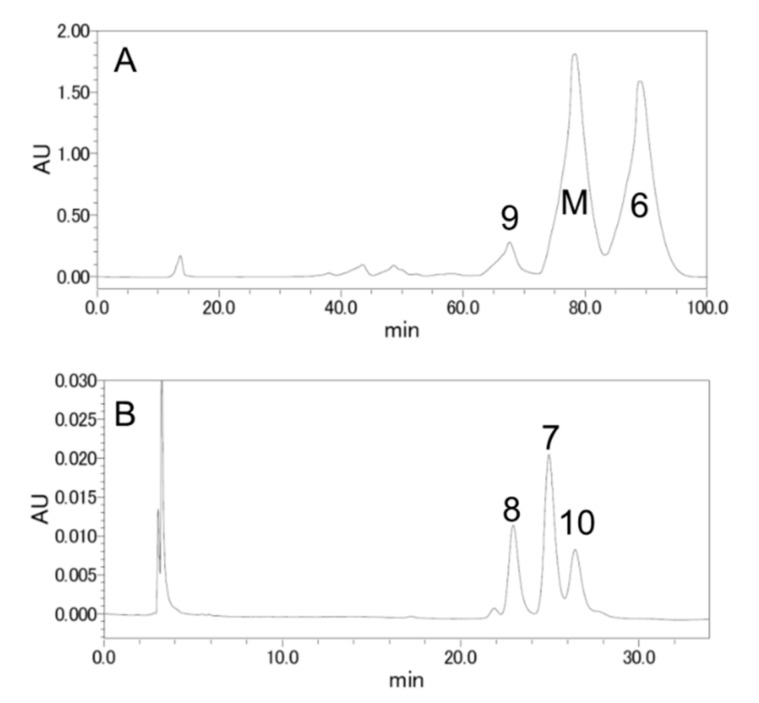
HPLC profile of target compounds **6**–**9** and compound **10**. After final reaction with 0.1% BHA in cyclohexane under reflux conditions for 1 h, the reaction mixture was concentrated. The compound which has the same Rf value (Rf = 0.4, AcOEt/*n*-Hex = 3:7) of TLC was purified using silica-gel column chromatography (AcOEt/*n*-Hex = 1:4). This compound was further purified with reverse-phase HPLC column chromatography, and the profile was shown in (**A)** (column: Mightysil RP-18 GP 250-20 (5 μm) Kanto Chemical Co., Inc, solvent: 100% acetonitrile; initially the flow rate was at 1 mL/min and it was increased to 5 mL/min over 1 min, and this condition was maintained, detector: 254 nm absorbance). As a result of NMR analyses, peak 9 (67.6 min) was attributed to compound **9** (vitamin D_2_), and peak 6 (88.9 min) was attributed to compound **6** (vitamin D_5_). Unfortunately, peak M (78.4 min) was a mixture. Peak M was further purified using a special HPLC column, and the profile was shown in (**B**) (column: COSMOSIL Cholester 4.6 × 250 mm Nacalai Tesque Inc, solvent: 100% acetonitrile; 1 mL/min, detector: 254 nm absorbance). As a result of NMR analyses, peak 8 (22.9 min) was attributed to compound **8** (vitamin D_6_), peak 7 (24.9 min) was attributed to compound **7** (vitamin D_7_), and peak 10 (26.4 min) was attributed to vitamin D_4_ (Figure 4. Compound **10**).

**Figure 4 metabolites-09-00107-f004:**
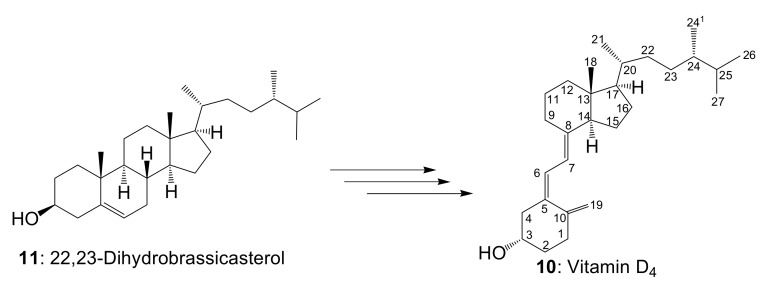
Predicted synthetic pathway of vitamin D_4_ (**10**). As a result of the synthesis of vitamin D_4_ (**10**), it believed that 22,23-dihydrobrassicasterol (**11**) was contained in the starting compound, commercially available β-sitosterol (**1**).

**Figure 5 metabolites-09-00107-f005:**
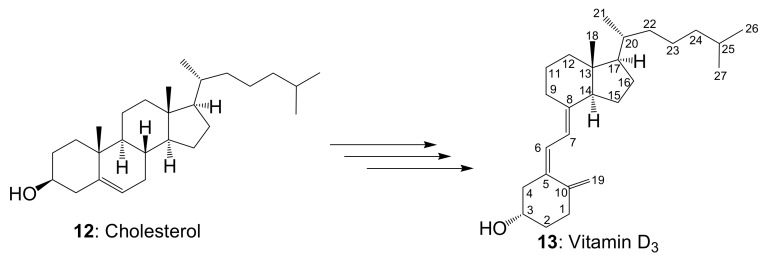
Vitamin D_3_ synthesis for comparative analysis by NMR. Synthesis of vitamin D_3_ (**13**) from cholesterol (**12**) was carried out in the same manner as the synthesis shown above. However, since pure cholesterol was used as a starting compound, highly pure vitamin D_3_ could be synthesized by silica-gel column chromatography purification.

**Figure 6 metabolites-09-00107-f006:**
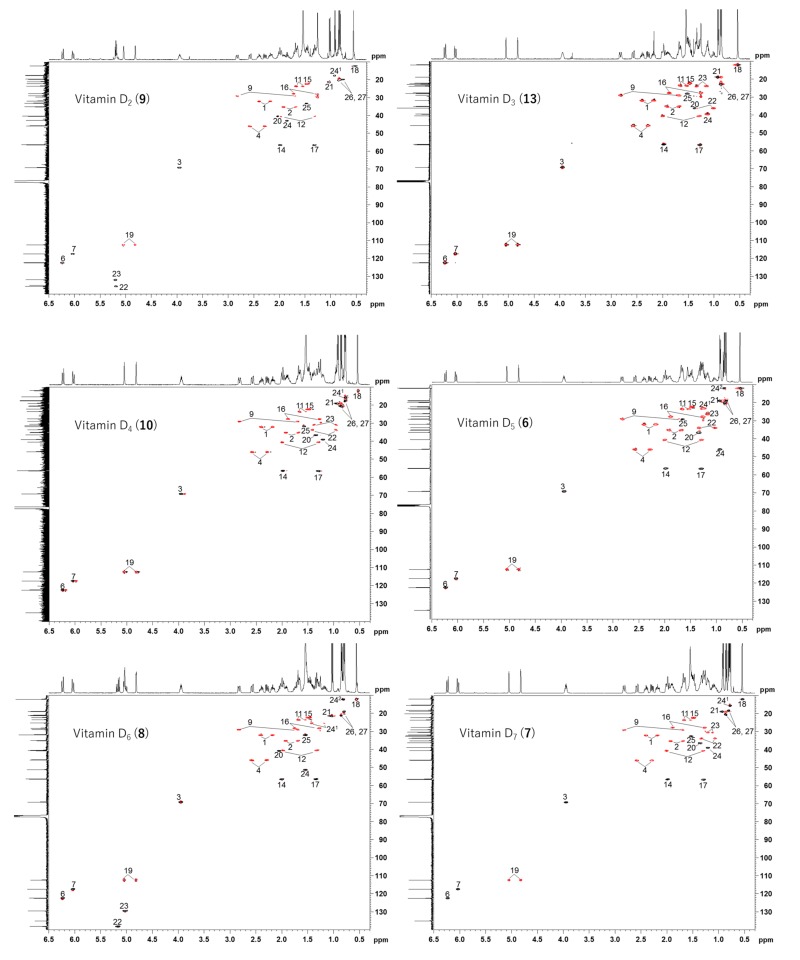
HSQC NMR spectra of vitamins D_2_ (**9**), D_3_ (**13**), D_4_ (**10**), D_5_ (**6**), D_6_ (**8**), and D_7_ (**7**). Methine (CH)/methyl (CH_3_) resonances have a positive intensity and are plotted in black; methylene (CH_2_) resonances have a negative intensity and are plotted in red.

**Figure 7 metabolites-09-00107-f007:**
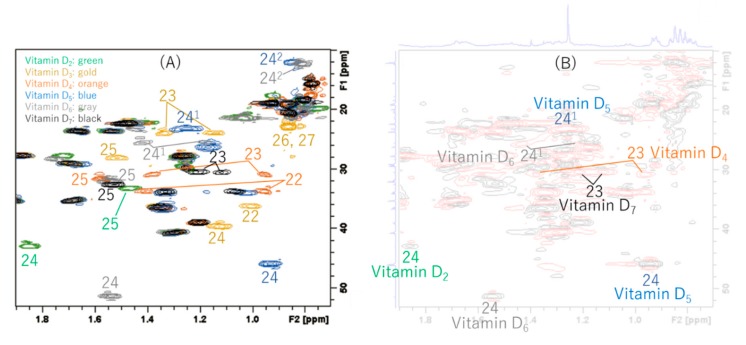
Analyses of HSQC NMR spectra. (**A**) Superimposed HSQC NMR charts of vitamins D_2_, D_3_, D_4_, D_5_, D_6_, and D_7_ were measured separately. They are color coded in green, gold, orange, blue, gray, and black, respectively. The following characteristic peaks were observed respectively. Vitamin D_2_ has CH-24, 25, vitamin D_3_ has CH-22, 23, 24, 25, 26, 27, vitamin D_4_ has CH-22, 23, 25, vitamin D_5_ has CH-24, 24^1^, 24^2^, vitamin D_6_ has CH-24, 24^1^, 24^2^, 25, and vitamin D_7_ has CH-23, 25. Although not in this range, both vitamins D_2_ and D_6_ also have CH-22, 23 as characteristic peaks. (**B**) HSQC NMR of the vitamin D mixture synthesized in this study. Methine (CH)/methyl (CH_3_) resonances have a positive intensity and are plotted in black; methylene (CH_2_) resonances have a negative intensity and are plotted in red. This mixture was obtained using silica-gel column chromatography before HPLC separations. The following characteristic peaks were observed respectively. Vitamin D_2_ has CH-24, vitamin D_3_ has no characteristic peaks, vitamin D_4_ has CH-23, vitamin D_5_ has CH-24, 24^1^, vitamin D_6_ has CH-24, 24^1^, and vitamin D_7_ has CH-23. From this analysis, it was found that vitamins D_2_, D_4_, D_5_, D_6_, and D_7_ were contained and vitamin D_3_ was not contained in the mixture.

**Table 1 metabolites-09-00107-t001:** Carbon-13 chemical shifts (*δ* ppm) of the vitamin D family.

Carbon	Vitamin D_2_ (9)	Vitamin D_3_ (13)	Vitamin D_4_ (10)	Vitamin D_5_ (6)	Vitamin D_6_ (8)	Vitamin D_7_ (7)
1	31.9	31.9	31.9	31.9	31.9	31.9
2	35.2	35.2	35.2	35.2	35.2	35.2
3	69.2	69.2	69.2	69.2	69.2	69.2
4	45.9	45.86 or 45.93	45.85 or 45.93	45.84 or 45.86 or 45.93	45.9	45.86 or 45.93
5	135.0	135.0	135.0	135.0	135.1	135.0
6	122.5	122.5	122.5	122.5	122.5	122.5
7	117.5	117.5	117.5	117.5	117.5	117.5
8	142.2	142.4	142.4	142.3	142.2	142.4
9	29.9	29.0	29.0	29.0 or 29.2	29.0	29.0
10	145.1	145.1	145.1	145.1	145.1	145.1
11	23.6	23.6	23.6	23.6	23.6	23.6
12	40.37 or 40.41	40.5	40.5	40.5	40.4	40.5
13	45.8	45.86 or 45.93	45.85 or 45.93	45.84 or 45.86 or 45.93	45.8	45.86 or 45.93
14	56.4 or 56.5	56.4	56.3	56.4	56.5	56.4
15	22.2	22.3	22.3	22.3	22.3	22.3
16	27.8	27.6	27.6	27.7	28.2	27.7
17	56.4 or 56.5	56.6	56.5	56.5	56.4	56.6
18	12.3	12.0	12.0	12.0	12.2	12.0
19	112.4	112.4	112.4	112.4	112.4	112.4
20	40.37 or 40.41	36.1	36.5	36.5	40.7	36.2
21	21.1	18.8	19.0	18.9	21.3	18.8
22	135.6	36.1	33.7	33.9	138.1	33.7
23	132.0	23.9	30.6	26.1	129.5	30.3
24	42.8	39.5	39.1	45.84 or 45.86 or 45.93	51.2	38.8
24^1^	17.6	-	15.4	23.1	25.4	15.4
24^2^	-	-	-	12.0	12.2	-
25	33.1	28.0	31.5	29.0 or 29.2	31.9	32.4
26	19.6 or 19.9	22.5 or 22.8	17.6 or 20.5	19.0 or 19.8	19.0 or 21.1	18.2 or 20.2
27	19.6 or 19.9	22.5 or 22.8	17.6 or 20.5	19.0 or 19.8	19.0 or 21.1	18.2 or 20.2

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
