# Peer review of "Simultaneous Synthesis of Vitamins D2, D4, D5, D6, and D7 from Commercially Available Phytosterol, β-Sitosterol, and Identification of Each Vitamin D by HSQC NMR"

_metabolites, 2019, doi:10.3390/metabo9060107_

Round 1

Reviewer 1 Report

In this manuscript the authors reported synthesis of vitamin D congeners from a commercially available beta-sitosterol (purity >40%), a mixture of, at least, four phytosterols considering the results that final products contained vitamin D2, D4, D5, D6, and D7.  Before publication, the following points require attentions:

1) Content of phytosterol congeners should be determined, both in molar ratio and in weight. Although the authors calculated each yield (Figure 2 and experimental section) based upon the beta-sitosterol, phytosterol congeners (beta-sitosterol, campesterol, stigmasterol, brassicasterol, and its reduced compound) have different molecular weights. In particular, the yields indicated for the final products were all unclear.

2) In page 6, lines 215-216, neither in experimental section, the recovery yield and the amount of previtamin D (compound 5) were not reported.

3) Although the authors referred the synthesis of vitamin D3 from cholesterol in the similar conditions, the experimental details and schemes or figures are missed.

Author Response

Our paper, originally submitted to journal of “Nutrients”, has been changed to journal of “Metabolites”. Therefore, the format of the paper is changed as follows.
2. Materials and Methods
3. Results and Discussion

2. Results and Discussion
3. Materials and Methods

(Reviewer 1)

In this manuscript the authors reported synthesis of vitamin D congeners from a commercially available beta-sitosterol (purity >40%), a mixture of, at least, four phytosterols considering the results that final products contained vitamin D2, D4, D5, D6, and D7.  Before publication, the following points require attentions:

1) Content of phytosterol congeners should be determined, both in molar ratio and in weight. Although the authors calculated each yield (Figure 2 and experimental section) based upon the beta-sitosterol, phytosterol congeners (beta-sitosterol, campesterol, stigmasterol, brassicasterol, and its reduced compound) have different molecular weights. In particular, the yields indicated for the final products were all unclear.

The following sentences has been added.
Line 97: The yield of the final step was 39%. It was calculated from the total weight (20.8 mg) of the obtained vitamin D analogs, including vitamin D4. Compound 5 (53.31 mg) was reacted to obtain vitamin D2 (9: 1.75 mg, 4.41
μmol), vitamin D4 (10: 1.85 mg, 4.64 μmol), vitamin D5 (6: 11.27 mg, 27.3 μmol), vitamin D6 (8: 2.41 mg, 5.87 μmol), and vitamin D7 (7: 3.52 mg, 8.83 μmol). Assuming that the ratio of brassicasterol/22,23-dihydrobrassicasterol/b-sitosterol/stigmasterol/campesterol contained in commercially available b-sitosterol does not change in each reaction, the weight percentage of each compound in commercially available b-sitosterol is 1.75 : 1.85 : 11.27 : 2.41 : 3.52 = 8.4 wt%/8.9 wt%/54.2 wt%/11.6 wt%/16.9 wt%, respectively. The molar ratio is 4.41 : 4.64 : 27.3 : 5.87 : 8.83 = 8.6%/9.1%/53.5%/11.5%/17.3%, respectively.

Description of Figure 2 was changed.

Line 132 (e) 0.1% BHA in cyclohexane, reflux, 1 h, 6 (vitamin D5): 21.1%, 7 (vitamin D7): 6.6%, 8 (vitamin D6): 4.5%, 9 (vitamin D2): 3.3%. → (e) 0.1% BHA in cyclohexane, reflux, 1 h, 39% {It was calculated from the total weight (20.8 mg) of the obtained vitamin D analogs, including vitamin D4.}.

2) In page 6, lines 215-216, neither in experimental section, the recovery yield and the amount of previtamin D (compound 5) were not reported.

The following sentence has been added.

Line 66: At this time, previtamin D (5) was not recovered.

3) Although the authors referred the synthesis of vitamin D3 from cholesterol in the similar conditions, the experimental details and schemes or figures are missed.

The following part has been added.

Line 173: (Figure 5; Figure S1 and the section of “The experimental details on the synthesis of vitamin D3 (13)” in supplementary materials).

In supplementary materials: pages S2 – S3 were added.

Reviewer 2 Report

The authors describe an interesting approach for a comparative NMR study where numerous vitamin D analogs (vitamins D2, D3, D4, D5, D6, and D7) are synthesized using same sources (i.e.phytosterols). The main question was accordingly addressed.

Although methods for synthesizing separate vitamin D analogs from phytosterols have already been done, the authors claim that methods for obtaining vitamin D analogs simultaneously have not been developed yet, what seems to be true.

However, by contrast to vitamin D2 and D3, vitamins D4, D5, D6, and D7 do not seem to play an important role for fortification purposes or other applications, and the authors have not mentioned why these analogs are of interest for them or for the reader. Vitamins D2 and D3 are readily commercially available and their synthesis is well established in an industrial scale. Particularly, is not obvious/understandable why the authors have repeated a well known synthesis of Vitamin D3 from cholesterol for their comparative NMR analysis study.

The text is well written and easy to read. Experimental details are given, so that the reader is able to follow the procedures and would also be able to reproduce the experiments.

Conclusions are consistent in terms that characteristic peaks of each vitamin D analog are identified, what could be useful for other researchers active in this field. The unique approach revealed some incorrect assignments of NMR chemical shifts in the literature that was overseen before in cases where each vitamin D analog was synthesized separately.

Analytical data mentioned in the Materials and Methods section should be shifted in the Results and Discussion section.

The last sentence of the conclusion should be rewritten.

Author Response

(Reviewer 2)

The authors describe an interesting approach for a comparative NMR study where numerous vitamin D analogs (vitamins D2, D3, D4, D5, D6, and D7) are synthesized using same sources (i.e.phytosterols). The main question was accordingly addressed.

Although methods for synthesizing separate vitamin D analogs from phytosterols have already been done, the authors claim that methods for obtaining vitamin D analogs simultaneously have not been developed yet, what seems to be true.

However, by contrast to vitamin D2 and D3, vitamins D4, D5, D6, and D7 do not seem to play an important role for fortification purposes or other applications, and the authors have not mentioned why these analogs are of interest for them or for the reader.

The following sentences has been added.

Line 32: In addition, active form of vitamin D3 are used as anti-osteoporosis drug, but hypercalcemia due to excessive intake has been reported as a side effect. Active forms of vitamins D4 and D7 have been reported to be effective in improving this side effect[4]. Therefore, it is very important to compare each vitamin D to examine its functions.

ref 4: Tachibana, Y.; Yokoyama, S.; Tejima, T.; Okamoto, Y.; Hongyo, T. 1 alpha-hydroxy vitamins D7 and D4' processes for the preparation thereof and pharmaceutical compositions. EP0562497(A1), 1993.

Vitamins D2 and D3 are readily commercially available and their synthesis is well established in an industrial scale. Particularly, is not obvious/understandable why the authors have repeated a well known synthesis of Vitamin D3 from cholesterol for their comparative NMR analysis study.

The following sentences has been added.

Line 174: Vitamin D3 is commercially available and does not need to be synthesized this time. However, in order to reconfirm the effectiveness of this synthetic method, it was intentionally synthesized from cholesterol.

The text is well written and easy to read. Experimental details are given, so that the reader is able to follow the procedures and would also be able to reproduce the experiments.

Conclusions are consistent in terms that characteristic peaks of each vitamin D analog are identified, what could be useful for other researchers active in this field. The unique approach revealed some incorrect assignments of NMR chemical shifts in the literature that was overseen before in cases where each vitamin D analog was synthesized separately.

Analytical data mentioned in the Materials and Methods section should be shifted in the Results and Discussion section.

The order of the Materials and Methods section and the Results and Discussion section was reversed. And the Figures 2, 3, 4, and 5 were moved to the Results and Discussion section.

The last sentence of the conclusion should be rewritten.

Line 360: These data will be used to enable comprehensive NMR analyses of vitamin D analogs, and it is expected that vitamin D research will be further advanced because of this.

These data will be used to enable comprehensive NMR analyses of vitamin D analogs, and the compounds synthesized here will be used to elucidate the functions of each vitamin D analog in our future studies.

Round 2

Reviewer 1 Report

 I have checked the manuscript. I confirm that the paper has improved to be acceptable to the journal.